# Understanding the mechanisms of infodemics: Equation-based vs. agent-based models

**Cristian Berceanu[1], Francesco Bertolotti[2], Nadia Arshad[3,4], Monica Patrascu**[1,5]*

**1** Complex Systems Laboratory, Department of Automatic Control and Systems Engineering, University Politehnica of Bucharest, Bucharest, Romania, **2** Intelligence, Complexity, and Technology Lab (ICT Lab) and School of Industrial Engineering, LIUC Università Cattaneo, Castellanza, Italy, **3** Section for Epidemiology and Medical Statistics, Department of Global Public Health and Primary Care, University of Bergen, Bergen, Norway, **4** Department of Clinical Neuroscience, Karolinska Institute, Stockholm, Sweden, **5** Centre for Elderly and Nursing Home Medicine and Neuro-SysMed Center, Department of Global Public Health and Primary Care, University of Bergen, Bergen, Norway

* monica.patrascu@uib.no

## Abstract

In an era where digital communication accelerates the global spread of false narratives, understanding how misinformation and disinformation propagate, especially during crises such as the COVID-19 pandemic, is vital to public health and policy. To delve into the diffusion mechanisms of misinformation (unintentionally false information) and disinformation (intentionally false information), we introduce a novel enhanced agent-based model (ABM) that integrates psycho-social factors and communication networks, which are elements often overlooked in traditional equation-based models (EBMs). We assess the two distinct techniques (ABMs and EBMs) through the lens of six classical SEIRS-class models (S susceptible, E exposed, I infected, R recovered). Beside the enhanced ABM, we also develop a simple ABM to emulate the EBM structure. We compare the ABMs with the EBMs over their entire parameter ranges in a total of 11110 experiments. Results show an overall weak equivalence between the two types of models, even if, under certain conditions, the outcomes of the EBMs and ABMs are similar. Furthermore, we evaluate the two model types by fitting them to real-world infodemic data on vaccine acceptance over 36 weeks using a multi-objective optimization procedure. The enhanced ABM shows an exceptionally better fit to real-world data (Pearson's correlation coefficient $\rho = 0.872$ and normalized root mean of square error NRMSE = 0.055) than the EBM ($\rho = -0.067$, NRMSE = 0.418) and the simple ABM ($\rho = 0.391$, NRMSE = 0.103). These findings underscore the critical role of model structure in capturing infodemic dynamics, and advocate for the use of ABMs when psycho-social influences and network interactions are central to the phenomenon.

**Data availability statement:** Models, experiment configurations, and heatmap results are available in the Github Repository "Understanding the Mechanisms of Infodemics: Equation-Based vs Agent-Based Models" (https://github.com/cristian-berceanu/ understanding_the_mechanisms_of_ infodemics). The real world data is described by Collis, A., Garimella, K., Moehring, A. et al. Global survey on COVID-19 beliefs, behaviours and norms. Nat Hum Behav 6, 1310–1317 (2022). https://doi.org/10.1038/s41562-022-01347-1, and can be accessed via the dataset "COVID-19 Beliefs, Behaviors & Norms Survey" (https://covidsurvey.mit.edu/).

**Funding:** No direct funding is declared for this article. MP would like to thank the GC Rieber Foundation, the Norwegian Government, and the Research Council of Norway (288164) for supporting their work at the Centre for Elderly and Nursing Home Medicine, University of Bergen, Norway. The funders had no role in study design, data collection and analysis, decision to publish, or preparation of the manuscript.

**Competing interests:** The authors have declared that no competing interests exist.

## Introduction

In 1995, a year that highlights the rising wave of internet-related development, Kevin Kelly was writing [1]: "The central act of the coming era is to connect everything to everything." Nearly 30 years later, we are witnessing an unprecedented upsurge in internet-based social media platforms and instruments for person-to-person and person-to-group communication. Humans, more so than before, are now connected in ways that foster high and ubiquitous access to information and to each other [2]. Our social networks are larger than in previous decades, in both number of actors (nodes) and ties (edges), which makes way for new questions and calls for new analysis methodologies [3]. The field of complex systems provides such opportunities in the form of network science, simulation of computational models, and specific concepts (e.g., small worlds, co-evolution, emergence, etc.) [4,5].

Social communities can be considered multi-layered networks in which people interact through a variety of relationships. This interpretation allows them to be investigated as complex networks [6,7]. When considering the available communication channels (phone, messaging apps, social media platforms, etc.), the complexity of these systems increases, both in size and relationship heterogeneity [8]. Topology-based modeling is already intricate [9], but when we must also account for the content of the information exchange between nodes, modeling social networks overlaid onto social media or communication networks becomes even more challenging. In a complex systems perspective, humans can be seen as systems with agency (capable of autonomous decision-making and reasoning), while the interactions between them are interpreted as signals: either information, energy, or matter over time. For the specific case of information diffusion, this approach allows us to consider the exchanged content.

In complex systems science, the chosen level of analysis typically depends on the modeling purpose and can significantly impact the overall results [10,11]. To enhance scientific communication and ensure replicability, researchers often employ a simplifying interpretative method, which results into a model on one of three scales [12]: a) macroscopic, when the whole network is represented as a black-box (i.e., without understanding its internal behavior), and only large-scale variations of outcomes are observed; b) mesoscopic, in which some parts of the network are differentiated, but the level of granularity does not permit modeling individual nodes; and c) microscopic, when each entity in the network has agency and manages its own dynamics and psycho-social interactions.

Given this framework, the study of social-network information diffusion on a microscopic level is of increasing interest. Network science has been investigating the emergence of social ties and how ideas could travel through internet-enabled media, such as political blog analysis [13] and election manipulation [14]. During the COVID-19 pandemic, the diffusion of untrue or malicious information resulted in an infodemic [15], and a co-evolution effect has been observed with the spread of the virus itself [16]. Consequently, analyzing and predicting the effect of infodemics is of high interest [17], and efforts have been focused on integrating epidemiological models of viral spread with those of information diffusion [16,18].

## Study aim and hypothesis

The aim of this paper is to investigate two model types for the spread of mis- and disinformation: macroscopic equation-based models (EBMs) and microscopic agent-based models (ABMs). We hypothesize that microscopic agent-based models are necessary to capture the elements of the human psycho-social context. To achieve this, we define a set of key objectives:

1. Model development based on six classical viral spread EBMs that comprise the SEIRS-class, with S susceptible, E exposed, I infected, R recovered:
   - Develop a simple ABM parameterized to entail mis- and disinformation behaviors at microscopic level, which emulates the assumptions of the classical EBMs.
   - Develop an enhanced ABM that, aside from the simple ABM functionality, accounts for the communication network and psycho-social interactions.
2. Inter-model equivalence of EBMs and ABMs:
   - Perform a detailed cross-correlation analysis over the entire parameter range for all models
   - Compare the models via a fitting experiment using real-world infodemic data

## Background

### Information diffusion models

A diffusion model is a framework, usually implemented mathematically or computationally, that represents the spreading of entities or features in a space [19], which can be either topological or relational (i.e., a network) [20]. Even if they were first developed in physical sciences, social sciences have widely adopted them, assuming that different kinds of features could spread within a population [21]. Diffusion models provide a systematic way to understand and analyze the dynamics of spread processes as a means of predicting their outcomes or investigating the factors that influence diffusion [22]. Diffusion models are well established in different disciplines, such as epidemics [23,24], information spread [25,26], opinion and strategy dynamics [27], and the economics of innovation [28,29]. In particular, the epidemic models derived from Kermack and McKendrick's seminal works [30–32] have received increased attention. The underlying assumptions of these models are: a) an entity within a population could exist in one specific state (i.e., susceptible, infected, recovered, or exposed); b) an infection could be transmitted by proximity to another individual (i.e., in the classic susceptible-infected-recovered SIR model the number of newly infected individuals at time *t* depends on the product of susceptible and infected individuals, which stands for the number of interactions); and c) any state could be reached by internal dynamics. Infodemic models equate "infection" with "knowledge", thus proposing an analogous interpretation [33].

Diffusion models are generally characterized by four elements: the structure of the interactions, dynamics of the interactions, possible states that model the diffused feature, and their dynamics. A classical approach to the diffusion model (for instance, the one adopted by Kermack and McKendrick [30] which is derived from Newtonian mechanics), consists of modeling a population as state variables. Each state typically varies over time according to a differential equation, such that the structure of the model is an oriented graph where each node is a population subset in a given state, and the connections depict functional dependencies. This approach is often called equation-based modeling [34,35]. In contrast, in the last 30 years, an opposite paradigm has emerged, where the atomic unit of the model is not the state but the individual decision-making entity itself [36], usually represented as a computational object [37]. When these computational entities are agents (i.e., systems with agency [38]), the methodology is known as agent-based modeling [39,40].

In spread modeling, equation- and agent-based approaches are often seen as concurrent, with complementary strengths and weaknesses [41,42], and there has been a long discussion regarding the preference for each of these methods [43], as well as the effect of the interaction structure on diffusion over a network [21,25] or more generally on its dynamics [44]. On the one hand, EBMs are much less computationally expensive, at the cost of assuming homogeneity of

sub-populations and using a mean-field approximation to the structure of interactions [45]. On the other hand, the smaller the population, the more individual features matter, and employing ABMs adheres closer to reality [46,47].

In epidemiology, there is an observable interest in assessing which methodology is better [48–52] and in identifying strategies to develop hybrid models that encompass the best of both [46,53,54]. However, very few studies establish a correlation between epidemics and infodemics [55].

## Infodemics

According to the World Health Organization, "an infodemic is too much information including false or misleading information in digital and physical environments during a disease outbreak" [56]. Based on the degree of misleading intentionality, diffused untruths can be classified into [57,58]: a) misinformation, which spreads without the intention to mislead; b) disinformation, which is produced purposely to cause harm (e.g., reporting manipulated statistics); c) malinformation, which refers to broadcasting true information to cause harm, such as circulating a report without its original context; d) rumor, which concerns the distribution of unverified information; and e) fake news, which are fabricated information mimicking news content.

On the one hand, gossip and rumor-sharing contribute to building and maintaining social ties. The drive to express views and perspectives in unreliable circumstances fuels the spread of mis- and disinformation [59], which has had detrimental effects on public health and politico-economic issues [60]. Notably, the waves of misinformation associated with the COVID-19 pandemic have raised multiple psychological and psycho-social issues that led to inappropriate measures, political instability, and mistrust in governing bodies [61–64]. Even when scientific inaccuracies are confined within seemingly closed communities [65], they have widespread and destructive effects on social groups, especially marginalized ones, leading to widescale societal changes and unrest [66].

On the other hand, misinformation and disinformation affect reliability and trust in social media networks as avenues for the dissemination of news or other important verified facts. In general, the spread of untruths intended to manipulate the perceptions of users has been recognized as a fundamental issue in democratic societies [67]. Subsequently, detecting falsity in diffused information has become an important concern, and studies are looking to minimize affected users and reduce propagation [68].

We surmise that proper analysis tools and models are necessary, to study not only the paths of misinformation but also its long-term impact on human psycho-social behavior.

## Methods

In this section we describe the equation- and agent-based models for mis- and disinformation diffusion, the comparative analysis method, and tools for model implementation and development. All models and results are available in the *Github* repository [69], including the scripts to create a docker image that runs the experiments for the two cases automatically.

### Preliminary analysis: A psychological perspective on the infodemic infection mechanisms

Infodemic models have been relying on the concept of "misinformation epidemic" [70], with subsequent deterministic models based on the epidemic SIR-class equations [42]. While this assumptions is not without merit, information travels in an inherently different manner than biological viruses, being affected by how communication is carried out and by individualized factors such as the illusion of knowledge [71].

Thus, in this section we analyze whether the four states of susceptible, exposed, infected and recovered are suitable for describing the spread of mis- and disinformation. Research on the psychology of misinformation has provided much insight into the psychological processes underlying susceptibility to misinformation in multiple domains [72,73]. Cognitive factors that contribute to supporting false beliefs include intuitive thinking (a lack of analytical thinking), cognitive failures (forgetting sources), and illusory truth (familiarity).

A recent model [74] proposes four psychological processes underlying susceptibility to health misinformation. Not only individuals with a capability to reason accurately are less susceptible, but also their resilience increases with the motivation to reason accurately. In contrast, directionally- and identity-motivated reasoning increases susceptibility, derived from a desire to reach a preferred conclusion that is often consistent with one's pre-existing views. These intertwining mechanisms are not possible to be explicitly included in EBMs, but they are suitable for ABMs.

The exposed state is that in which a piece of information (e.g., news, blog posts, communication from social contacts) takes a while to be processed, especially when its complexity makes it difficult to grasp, or requires repeated exposures to generate belief. However, the mechanism behind the establishment of false and accurate belief is the same [75]. People are often biased to believe in the validity of information and "go with their gut" and intuition when deciding what is true instead of deliberating [76]. For instance, 31% respondents in a U.S. survey ($n = 2023$) in March 2020 agreed that COVID-19 was purposefully created and spread, despite the absence of any plausible evidence for its intentional development [77]. People might have encountered conspiracy theories about the source of the virus many times, which might have contributed to this widespread belief because simply repeating a claim makes it more believable than presenting it only once [78]. Repetition increases belief in both misinformation and facts, and people get "infected". Regardless of cognitive ability and despite contradictory advice from an accurate source or accurate previous knowledge, there is a possibility that illusory truth persists months after the first exposure [79]. EBMs do not differentiate between repeated exposures, even though they do account for incubation time, whereas in an ABMs it would only be a matter of adding a condition to a state transition trigger. We thus put forward that the incubation rate should be redefined for infodemics.

But does belief really equate infection? [80] The exposure to false information is a strong contributor to the formation of false beliefs. Access to high-quality information, whether true or not, is not necessary; instead, a range of precursors, cognitive and socio-affective drivers, influence the formation and storage of false beliefs [75], which draws a parallel with the properties of cells that determine whether they are or are not hosts to viral multiplication. But misinformation and its siblings are not singular pieces that are passed on from person to person, nor are they easily identifiable by the "symptoms" of the "disease" they cause. Information and the knowledge it is stored as always come with related facts, untruths, beliefs, consequences, premises, and relationships that either support or restrain the "infection", thus affecting incubation, recovery, and loss of immunity. Where do we draw the line between infection and other beliefs that are not necessarily true but not harmful either? For instance, the belief that leaving scissors open on the table leads to discord is not true, but it serves the purpose of avoiding accidents caused by exposed blades. Perhaps the definition of the infected state in infodemic models needs more interdisciplinary investigation.

Even with these limitations, there are enough arguments to analyze infodemics based on the four epidemiological states. In what follows, we thus utilize the four *SEIR* states in the development of a simple ABM that emulates the corresponding EBMs, following the parallel with biological virus spreads. The psychological perspective paints a larger picture, for which we propose an enhanced ABM design that takes into account the properties of how information diffuses among humans. Based on this preliminary analysis, we select the agent state variables to encode social navigation (coordinates, orientation), the information they share with others (information status), and their cognitive engagement in communicating with others (energy). In terms of behavior, we choose to implement information-seeking based on access (amount in observable vicinity), affinity to peer groups (homophily), one-on-one exchange, and communication through different media (as information does not need physical contact between people to spread). The next sections describe in detail the mechanisms employed for the models, with a conceptual representation in Fig 1 and with pseudocode available in the supplementary material.

## Equation-based models (SEIRS)

The prevalent equation-based models (EBMs) for infodemics are based on epidemic interpretations of how information is transmitted throughout a population. They are macroscopic models, in which the persons engaged in receiving or relaying

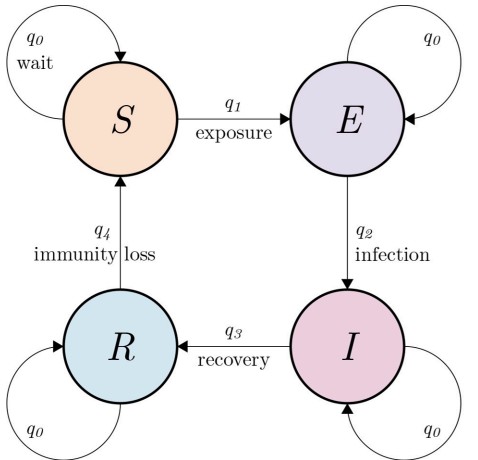
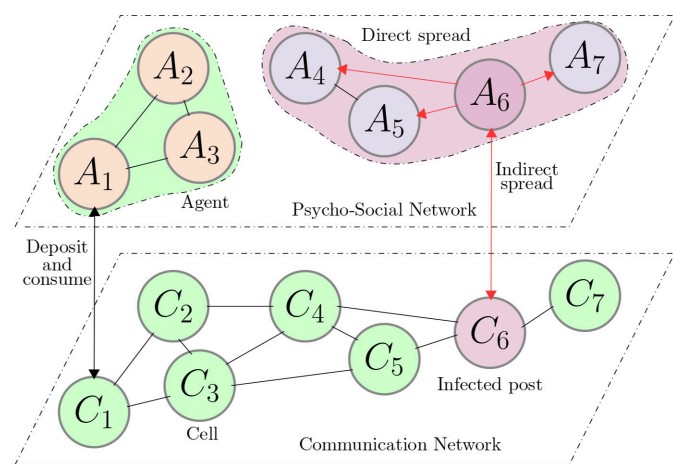

**(a)** *SEIRS* model representation with states ($S$ susceptible, $E$ exposed, $R$ recovered, $I$ infected) and transitions ($q_0$ wait for a trigger in current state, $q_1$ exposure, $q_2$ infection, $q_3$ recovery, $q_4$ immunity loss). All model combinations are listed in Appendix I.

**(b)** ABM structure (conceptual representation) in which the cells ($C_1 - C_7$) form the communication network, while the agents ($A_1 - A_7$) form the psycho-social network; the spread of mis- and disinformation can be direct (between agents) or indirect (through stigmergy).

**Fig 1**. **Conceptual representation of the two model types.** The equation-based model (*SEIRS*) is structured as a finite-state machine in which population subsets mass-transition between states based on probabilities. The agent-based model (ABM) is structured as layered complex networks in which individuals transition between states based on local interactions.

information lose their individuality. The main variables of the models become numbers reflecting portions of the population in one of several distinct states: susceptible, infected, recovered or exposed (Fig 1a). In this study, we analyze six such diffusion models: *SI*, *SIS*, *SIR*, *SIRS*, *SEIR* and *SEIRS* [42]. They are differentiated by how many states are defined within the population and how many state transitions are allowed. For instance, in the *SI* model, the states are susceptible and infected, with the only transition $S \rightarrow I$; the *SIS* model also allows the reverse transition $I \rightarrow S$. For brevity, we only describe model *SEIRS* in this section; all six models are listed in Appendix I.

*SEIRS* [81] consists of four equations as dynamic representations of the timewise state transitions of the four variables (Fig 1a), defined as shown in Eq (1).

$$\begin{cases} S'(t) = -\beta N^{-1} S(t) I(t) + \xi R(t), \\ E'(t) = \beta N^{-1} S(t) I(t) - \sigma E(t), \\ I'(t) = \sigma E(t) - \gamma I(t), \\ R'(t) = \gamma I(t) - \xi R(t). \end{cases} \tag{1}$$

Where $S(t)$, $E(t)$, $I(t)$, $R(t)$ are the numbers of individuals with susceptible, exposed, infected, or recovered state, respectively, at time $t$. Notations $S'(t), E'(t), I'(t), R'(t)$ represent the first order derivatives of these variables and they model how the states of the population change between two moments in time. $N$ is the total number of individuals in the population. The model parameters are: infection rate $\beta$, incubation rate $\sigma$, recovery rate $\gamma$, and immunity loss rate $\xi$.

Although *SEIRS*-class models can be expanded to include vital dynamics (asymmetrical birth and natural-cause death rates) [82], the total population is constant in the classic version: $N = S(t) + E(t) + I(t) + R(t)$.

The susceptible state is the nominal state of individuals and it models the population that can become either infected with (models *SI(S)* and *SIR(S)*) or exposed to (models *SEIR(S)*) mis/disinformation. The population with exposed state cannot infect other individuals, but can become infected after an incubation period. Those with infected state are able to

spread the "infection" to the susceptible individuals and can become recovered after a period of time. The individuals with recovered state are immune to the "infection" but can lose this immunity over time, at which point they transition to the susceptible state.

Parameter $\beta$ represents the infection rate and is defined as the population percentage switching states from susceptible to exposed (or infected) in the unit of time. Parameter $\sigma$ represents the incubation rate and is defined as the population percentage switching states from exposed to infected in the unit of time. Parameter $\gamma$ represents the recovery rate and is defined as the population percentage switching states from infected to recovered in the unit of time. Parameter $\xi$ represents the immunity loss rate and is defined as the population percentage switching states from recovered to susceptible in the unit of time.

For the *SEIRS* models, the unit of time is chosen to match the modeled epidemic, e.g. days for rapidly spreading infections, or months for slower dynamics. In this study, we consider the unit of time to be equivalent to the unit of time of the agent-based models, measured in *ticks*. Thus, we maintain the generality and scalability of the models.

## Agent-based models

Agent-based models are built with two types of entities: (a) an *agent* is a simulated system capable of perception, action, communication, and reasoning; (b) a *cell* is a simulated system part of the environment. The ABM unit of time is called a *tick*. Its real-world equivalence is selected based on problem context and dynamics. During one tick, all behaviors defined in the ABM (agents, cells, etc.) are executed in parallel asynchronously, by one step.

For this study, we propose an ABM structure to include the separation and dependence between the psycho-social component and the communication medium. We base this choice on the argument that one network alone is not sufficient to embody these types of dynamics, put forth by studies on epidemic-infodemic interactions [83] and opinion dynamics [84]. Thus, agents model individuals (who can interact with the environment or each other) and the psycho-social network, while cells model the communication channels and/or network (Fig 1b).

**Network structures.** The psycho-social network is formed of all agents in the ABM. From an information diffusion perspective, this network appears as a graph with temporary edges, which are formed when two agents meet to directly transmit information to each other. This reflects the real world in which two persons open ad-hoc point-to-point communication sessions (e.g., face-to-face, phone call) with each other, forming a connection, but are not actively transmitting information to each other around-the-clock.

The social media/communication network (further referred to as the environment) is formed of all the cells in the ABM. In this study, we interpret each cell as a platform (e.g., blogging, micro-blogging, forums, social apps), broadcasting entity (e.g., newspaper website, television) or communication channel (e.g., VoIP apps). Equivalent to the real world, an agent "navigates" among these platforms; in this paper, we visualize this process by allowing agents to travel across the cell grid. While the cells are connected via adjacency to each other, we do not consider the network edges relevant in this situation, as platforms themselves do not exchange information; instead, the agents carry it across media.

The environment thus represents the online (social) media network overlaid onto the direct point-to-point communication of the agents. The difference in behavior between direct communication and online media is that the latter functions as a broadcast instead of a conversation. Blog posts, news articles, etc. are stored online and available for anyone in the network. In the real world, this type of broadcast is not received by all individuals at the same time, as navigating through personal media dashboards limits exposure (e.g., local news relayed within a region and not internationally, or social media clusters formed of family members, etc.): we achieve this effect by implementing the *agent movement* functions as they seek interactions and limiting their perception to a radius around themselves.

**Information modeling.** The relationship humans have with information has been studied at length [85,86]. The interpretation we adhere to in this paper associates this interaction process with foraging behaviors, through which people seek, select, consume and avoid information [87,88]. For agents, cells are both information sources and sinks. Agents

generate information by "posting" on cells (interpreted as either new content, or interactions with existing content such as commenting). When agents are infected, they might deposit mis- or disinformation, which is then picked up by other agents. In complex systems, this process of communicating through the environment is known as stigmergy [89]. Agents gain energy by consuming information and lose energy by posting, thus simulating engagement (e.g., scrolling through dashboards, commenting) or exposure fatigue (e.g., avoiding news applets).

Aside from content deposited by agents on cells, we implement a generation mechanism through which information "grows". The underlying hypothesis is that information available in the environment can be generated by sources outside that environment. Thus, there is an external component to the model which agents cannot affect, but that can influence the agent or its group. For instance, the social cluster of an individual being influenced by the media, over which single individuals do not have, in turn, an effect. Moreover, if cells are instances of online media platforms where agents interact with content, the information they carry can be influenced by recommendation engine algorithms, which promote any type of engagement, whether positive or negative [90], and thus become vectors for mis- and disinformation.

We developed two models: (a) the *simple model* is a direct translation of the finite-state machine of the EBMs into agent behaviors in which misinformation spreads point-to-point between agents; (b) the *enhanced model* is built upon the simple model by adding psycho-social components to the agent behaviors (e.g., homophily), as well as the misinformation spread through the online (social) media network. All algorithms associated with the two models are detailed in Appendices II and III. The two models are briefly described in the following sections Simple ABM and Enhanced ABM using the ODD (Overview, Design Concepts and Details) Protocol [91].

**Simple ABM.**

**Overview:**

**The purpose** of the simple ABM is to model infodemics taking into account complex interactions between agents based on the information model.

**The state of an agent** $A_i$ ($i = 1..N$) is described by four variables: coordinates $(x_i, y_i)$, orientation $h_i$, information status $s_i$, and energy level $e_i$. Coordinates are discrete (integer) and spatially define the position of the agent in the environment. Orientation is defined by the heading angle. Information status is a categorical variable $s \in \{$susceptible, exposed, infected, recovered$\} = \{S,E,I,R\}$ with four categories equivalent to the SEIRS model. Energy level is a continuous variable modeling the interest of agents to communicate (receive and relay information) and thus to move through the environment.

**The state of a cell** $C_j$ ($j = 1..M^2$) is described by two variables: coordinates $(x_j, y_j)$ and information quantity $q_j$. Cells are static, arranged on a torus, and visible in the model interface as an $M \times M$ square grid (agents leaving one border reappear on the opposite side). Together, cells form the communication network. Coordinates are discrete (integer) and define the position of the cell inside the grid. Information quantity is a continuous variable representing how much information is available to agents in one particular communication channel (displayed in shades of green: lighter for less information, darker for more). For this model all available information is considered true.

Fig 2 shows the initialization of the simple ABM at $t = 0$ *ticks* and the visualization of the model at $t = 50$ *ticks*.

**The scheduler** triggers the two main behaviors of the agent: *Agent movement* and *Agent information status*. The *agent information status* describes how the agent's state changes as a consequence of direct (point-to-point communication) and indirect (information posting and consumption) diffusion of information. The implementation of the scheduler is explained in Appendix II.

**Design concepts:**

**Agent movement.** Agents move through the environment toward a new position with a specified heading. The agent's target is computed based on vicinity: choosing the cell (patch) with the highest amount of information in agent radius $r$ and viewing angle $\theta$. The meeting of two agents on the same cell represents two individuals simultaneously using the same instance of a communication channel (e.g., a phone call, a messaging app, face-to-face conversation, etc.).

**Agent information status.** The categorical agent state variable $s \in \{S,E,I,R\}$ defines the set of actions it can perform, i.e. its behaviors (algorithms in Appendix III). The transitions between the four categories follow the rules of the EBMs

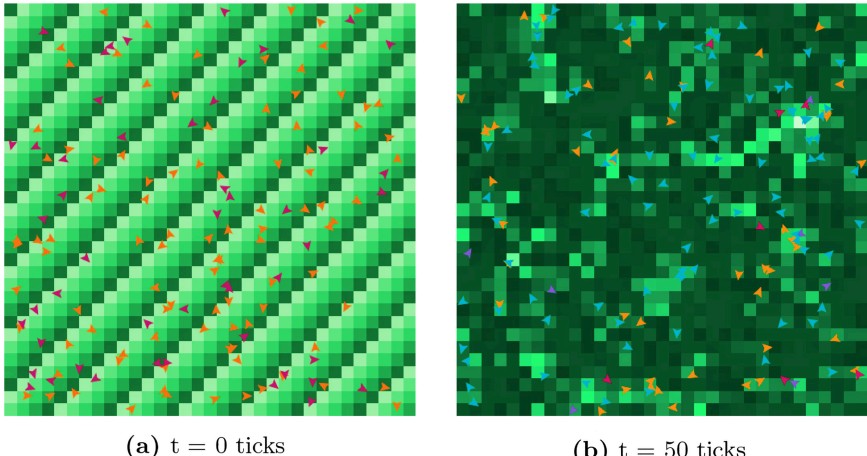

**(a)** t = 0 ticks      **(b)** t = 50 ticks

**Fig 2**. **Visual representation of the agent-based implementation of the diffusion model at two different time steps.** The color of each cell (shades of green) represents the level of information present in that specific area, while agents are depicted by arrows in a visual representation that shows both the position and the current orientation: orange susceptible, magenta infected, blue recovered, and violet exposed.

finite-state machine and are implemented via threshold tests with the probabilities of infection $\beta$ (relative to world and population size), successful incubation $\sigma$, recovery $\gamma$, and immunity loss $\xi$.

The simple model implements diffusion using point-to-point communication between agents, meaning that mis- and disinformation can only be transmitted when agents talk directly to each other. The point-to-point communication is established by infected agents with closest susceptible agents. This is the epidemiological interpretation in which biological viruses are transmitted through direct contact. Agents lose energy by moving through the world. This ABM is designed to allow for disinterest or sudden interest, and thus implement vital dynamics (e.g., "death" by leaving the network or "birth" by joining the network). However, to match the ABM to the classical SEIRS EBMs, we deactivated these functions. Because in this case the environment does not provide or store false information, it does not affect the dynamics of the infection spread, thus allowing for a fair comparison between the two modeling approaches.

While the movement path of agents is *random*, the starting location of all agents is *deterministic*. The agents *sense* their environment and *interact* with other agents in their vicinity ($d(A_i, A_j) = \sqrt{(x_i - x_j)^2 + (y_i - y_j)^2} < r$) directly (point-to-point communication).

**Details:**

Each agent is initialized with a starting position, heading, and state (S,E,I, or R, depending on the submodel type). No data is exchanged with the outside world. We consider the following 6 submodels for the simple ABM: *SI*, *SIS*, *SIR*, *SIRS*, *SEIR* and *SEIRS*.

**Enhanced ABM.**

**Overview:**

**The purpose** of the enhanced ABM is to model infodemics taking into account complex interactions between agents based on the information model and communication network structure.

**The state of an agent** $A_i$ ($i = 1..N$) is described by five variables: coordinates, orientation, information status, energy level, and group $g_i$, which is a categorical variable that tags the agent as belonging to one of two social groups: $g \in \{G1, G2\}$.

**The state of a cell** $C_j$ ($j = 1..M$) is described by three variables: coordinates, information quantity, and information type $c_j$, which is a categorical variable $c \in \{c_{true}, c_{false}\}$. Truthful information $c_{true}$ is displayed as *green*, while falsity $c_{false}$ is displayed as *red*, both in shades based on quantity. Cells are arranged in the same grid as the simple model.

**The scheduler** triggers the main behaviors of the agent: *Agent movement and information status*. The *agent information status* describes how the agent's state changes as a consequence of direct (point-to-point communication) and indirect (information posting and consumption) diffusion of information. The implementation of the scheduler is explained in Appendix II.

**Design concepts:**

**Agent movement and information status.** Agent movement and information status are similar to the simple model. The difference is that the $S \to I$ and $S \to E$ transitions can be triggered externally by false information from the environment. The infection rate $\beta$ is also adjusted to account for two infection sources (other agents and cells).

**Homophily.** In this extension, we implemented a simple mechanic based on group adherence: agents in groups $G1$ and $G2$ will only believe false information from agents in the same group. Thus, the $S \to I$ and $S \to E$ transitions become conditional.

The enhanced model implements diffusion using both point-to-point and stigmergy communication between agents. Thus, mis- and disinformation can be transmitted when agents talk directly to each other and when they interact with the information shared through the communication network. Agents lose energy by moving through the world, but also when posting information. The point-to-point communication is similar to the simple model, while stigmergy allows for mis- and disinformation to spread through the shared media stored online, adding another informational dimension to the epidemiological interpretation.

While the movement path of agents is *random*, the starting location of all agents is *deterministic*. The agents *sense* their environment and *interact* with other agents in their vicinity ($d(A_i, A_j) = \sqrt{(x_i - x_j)^2 + (y_i - y_j)^2} < r$) directly (point-to-point communication) or indirectly (information posting and consumption) through the environment.

**Details:**

Each agent is initialized with a starting position, heading, and state (S,E,I, or R, depending on the submodel type). No data is exchanged with the outside world. We consider the following 6 submodels for the enhanced ABM: *SI, SIS, SIR, SIRS, SEIR* and *SEIRS*.

The differences in characteristics [92] between the EBM, the simple ABM, and the enhanced version of the ABM are summarized in Table 1.

**Implementation.** Both the simple and enhanced ABM function on the same base structure (scheduling), detailed in Appendix II together with the two model interfaces.

The scheduling of the model can be summarized into phases. The initialization phase loads the agent list and positions, and the cell distribution with information quantities into the model (here, we choose the same initialization for all model runs). The next phases execute agent behaviors.

First is the movement phase, in which an agent $A_i$ chooses a target to move toward. The decision-making mechanism considers a given radius $d^e$ around agent $A_i$. Agents deplete a fixed amount of energy $e_c$ for each movement. Second, the spreading phase, in which an agent $A_i$ with status $s_i = I$ spreads mis- or disinformation to the nearest neighbor $A_k$ with status $s_k = S$. To be infected or exposed, $A_k$ should be at a distance $d_{ik} < d^c$ (preset radius). Third, agent $A_k$ changes status to $s_k = E$ for models *SEIR(S)* or to $s_k = I$ for the others. Fourth, an agent $A_i$ with $s_i = I$ can recover to $s_i = R$ for *SIR(S)*, *SEIR(S)*. Model selection is possible through the ABM interface. Fifth, a recovered agent $A_i$ with $s_i = R$ can become susceptible in the case of models *SIS, SIRS, SEIRS*. (Deactivated function: an agent $A_i$ with negative energy $e_i$ is removed from the agent list, i.e., leaves the simulated society, which occurs whenever it expends more energy than it collects and stores.) The cell behavior execution phase consists of the information $q_j$ on each cell $C_j$ replenishing with a preset growth rate up to a maximum limit.

**Table 1**. Comparison of characteristics between the EBM, simple ABM, and enhanced ABM for infodemic modeling.

| Feature | EBM | Simple ABM | Enhanced ABM |
|---|---|---|---|
| Abstraction level | High (far from abstract biology) | Low (close to biology) | Low (close to biology) |
| Nature: analysis level (description level) | Synthetic: macroscopic (population-level) | Many parameters: microscopic (entity, individual-level) | Many parameters: microscopic (entity, individual-level) with network structure |
| Interactions (system homogeneity) | Homogeneous mixing (mean-field) | Point-to-point contact between agents | Heterogeneous; Point-to-point & indirect broadcast via cells; agents interact with media and each other |
| Information spread | Homogeneous population fractions; no heterogeneity of information | By agent contact, no explicit false/true informantion | By agent contact and via environment (cells hold true/false info); agents perceive and spread content |
| Modularity and incrementality | Little modular and incremental | Modular and incremental | Modular and incremental |
| Psycho-social aspects | Not included | Limited (energy/interest) | Includes social influence, psychological processes |
| Main structure and mathematical resolution | Formalized: differential equations (SEIRS-class) | Non-formalized: agents as individuals moving in a grid; direct interactions | Non-formalized: agents & communication cells; includes psycho-social and media networks |
| Runtime | Low | High | High |
| Computational requirements, scalability, flexibility | Low computation, high scalability and low flexibility | Moderate computation, moderate scalability and high flexibility | High computation (larger state space, more interactions), moderate scalability and high flexibility |
| States | S, E, I, R (at aggregate level) | S, E, I, R (at agent level) | S, E, I, R (at agent level |
| Ease of implementation | Moderate | Complex | Complex |
| Strengths | Analytical tractability, fast simulations, captures mean trends | Captures individual heterogeneity, emergence | Models network/media effects, higher realism |
| Limitations | Ignores individual/social structure, cannot capture emergence | No media/network effects, computationally intense, less realistic for infodemics | Computationally intensive, more complex calibration |

## Comparative analysis

Directly comparing EBMs and ABMs is crucial because these two approaches rely on fundamentally different assumptions about population structure and individual behavior. While EBMs offer analytical tractability by averaging over large populations, ABMs explicitly represent individual heterogeneity and local interactions, which can strongly influence diffusion dynamics but could be difficult to simulate when the number of agents exceeds a threshold. Thus, we compare pairs of homonymous outcome variables from the same-class model (e.g., the SI EBM's susceptible vs. the SI ABM's susceptible): evolution over time and scale. We focus on the key parameters common to the two frameworks ($\beta, \gamma, \xi, \sigma$), which we vary across their entire ranges. This approach allows us to highlight the conditions under which the two representations diverge.

To reduce the effect of randomness inherent to ABMs, we fix all initializations and the other ABM parameters. We thus define the same starting coordinates and orientation for each agent in the initial population, and specify the same initial distribution of information on cells and size of the homophily group. To account for probability-based choices (e.g., assigning an agent to a group), we apply the standard practice in complex systems simulation of repeating ABM experiments under the same initial conditions and averaging the set of outcomes before comparison with the EBM. In what concerns the implementation framework, NetLogo is deterministic in the sense that one simulation returns the same results every time on different devices, if it is initialized with the same seed (specifiable parameter).

The comparative analysis we perform in this study has two outcome measures which describe how well the ABM-generated variables $y_a \in \{S(t), E(t), I(t), R(t)\}$ match the corresponding EBM-generated variables $y_e$ over a predetermined period of simulated time, corresponding to a number of $q$ samples (i.e., *ticks* within the chosen period).

The Pearson correlation coefficient $\rho$ [93] is given by Eq (2).

$$\rho = \frac{\text{cov}(y_a, y_e)}{\sigma_{y_a} \sigma_{y_e}}. \tag{2}$$

Where $\sigma_{y_a}$ and $\sigma_{y_e}$ are the standard deviations. By calculating the sample-to-sample cross-sectional correlation between $y_a$ and $y_e$ over $q$ samples, we investigate whether this relationship is linear, which would mean that the transient responses of the models are similar over time. This measure thus shows how well the two variables match longitudinally (shape over time) and is defined over the interval $[-1; 1]$, where 1 is best match, $-1$ mirrored evolution, and 0 is complete mismatch.

The normalized root mean of square error (NRMSE) is defined as in Eq (3).

$$\text{NRMSE} = \frac{\sqrt{\frac{1}{q} \sum_{k=1}^{q} (y_{a_k} - y_{e_k})^2}}{y_{\max} - y_{\min}}. \tag{3}$$

Where $q$ is the number of samples, with $y_{\max}$ and $y_{\min}$ the maximum and minimum values of $y_a$ and $y_e$ (in this case, for a constant total population size, the same for both variables). NRMSE is a dimensionless score for scale-independent comparisons of model outcomes, defined over the interval $[0; 1]$, where 0 means best match and 1 worst [94].

Note that $\rho$ cannot be calculated when one of the contributing variables is zero; to avoid skewing the summary calculations, these were adjusted to match (1) or not-a-match (0) based on the NRMSE of the pair.

In this study, we use the combined outcome measures $\rho$ and NRMSE to assess the similarity of the model outcomes as discrete signals: $\rho$ quantifies how well $y_a$ and $y_e$ match in shape, whereas NRMSE offers an estimate of the relative differences between model outcomes. Together, the two metrics allow for an analysis of both transient and steady state responses of the models, while providing an intuitive descriptor of why the models do not match. For instance, the combination of $\rho = 0.9$ and NRMSE $= 0.8$ means that the models show a similar outcome variation over time, but with large differences in amplitude. In practical terms, this would mean that the two populations adapt in almost the same manner to the spread of misinformation, but the amount of people affected differs widely.

**Fitting to real-world data: Case study on vaccine acceptance**

We also evaluate the two types of models in a fitting experiment. The real world data we choose for this case study describes vaccine acceptance and disapproval during the pandemic. The model type is SIS, in which we consider the susceptible state as "accepting the vaccine" and the infected state as "not accepting the vaccine", with the possibility to recover to susceptible state.

**Infodemic real-world dataset.** The data was collected via a global survey on COVID-19 beliefs, described in [95] from 23 countries between June 2020 and March 2021. The data are available in the section Data Availability of the same paper [95]. One of the questions asked: "If a vaccine for COVID-19 becomes available, would you choose to get vaccinated?". Martinelli and Veltri [96] then conducted a study on COVID-19 vaccine acceptance, which produced a dataset expressing the percentage of the populating accepting of the vaccine. The dataset is longitudinal over 36 weeks, with a sample of 2 weeks between measurements. From this dataset, we choose Romania as a case study.

**Fitting method.** We search for the enhanced ABM and the EBM parameters so that the model outputs $S_i(k)$ and $I_i(k)$, with $i \in \{\text{ABM}, \text{EBM}\}$, match the real world data described by the real world susceptible population $S_{RW}(k)$ and the real world infected population $I_{RW}(k)$. Considering that the two signals are mirrored, for a parameter set $\pi_i$ and discrete time

step $k = 0..19$, the multi-objective optimization problem is given by Eq (4).

$$\min_{\pi_i} \left( \left. \rho(k) \right|_{S_i(k), S_{RW}(k)}, \left. \mathrm{NRMSE}^{-1}(k) \right|_{S_i(k), S_{RW}(k)} \right), k = 0..19. \tag{4}$$

The ABM parameter set is $\pi_{\mathrm{ABM}} = \{\beta, \gamma, P_M, T_F, I_R, L_M^{S,I}, L_P^{S,I}, D_P^{S,I}\}$, where $\beta \in [0, 1]$ is the infection rate and $\gamma \in [0, 1]$ is the recovery rate. $P_M \in [1, 15]$ is the population multiplier and scales the number of agents from a initialization baseline of 100. $T_F \in [1, 336] \cap \mathbb{N}$ is the time factor and determines the size of an ABM tick relative to the real world data sampling time; thus, $T_F$ represents how many ABM ticks pass for each week in the real world. $I_R$ is the information regrowth rate. $L_M^{S,I} \in [1, 100] \cap \mathbb{N}$ and $L_P^{S,I} \in [1, 100] \cap \mathbb{N}$ represent the energy loss of agents from movement and from posting information, for susceptible and infected agents, respectively. Finally, $D_P^{S,I} \in [1, 25] \cap \mathbb{N}$ represent the delay between agents consuming and posting information, for susceptible and infected agents, respectively.

The EBM parameter set is $\pi_{\mathrm{EBM}} = \{\beta, \gamma, T_F\}$, where $\beta \in [0, 1]$ is the infection rate and $\gamma \in [0, 1]$ is the recovery rate. $T_F$ represents how many data points we are considering for each week in the real world, $T_F \in [1, 336] \cap \mathbb{N}$.

For both, the population size is normalized to percentages and the two outcome variables of all models (susceptible and infected) are initialized to approximate the first datapoint of the real world data. To calculate the criteria $\rho$ and NRMSE, we bin the model outputs to the size of the real-world dataset (19 points) by averaging.

The optimization procedure is performed using the Python hyper-optimization library Optuna, which provides samplers and pruners and determines the importance of parameters and their interdependence. In this case study, we choose the Tree-structured Parzen Estimator (TPE) sampler and the hyperband pruner.

### Tools, frameworks, and software

The ABMs were implemented and simulated using NetLogo 6.3.0 [97]. The EBMs were implemented in Python 3.11.0 in discrete form, with a time step equivalent to one ABM tick. All visualizations, plots, heatmaps and calculations were also performed in Python 3.11.0. The following Python packages were used: matplotlib 3.7.1, numpy 1.24.3, scikit-learn 1.2.2, scipy 1.10.1, optuna 3.5.0, optuna-dashboard 0.14.0.

## Results

### Evaluation of intermodel equivalence

In this section we present a selection of relevant results (limited by space considerations), with more examples in Appendices IV and V, and a full set of simulation results in the repository. Results are generated from three instances of the ABMs: a) a small variant of the simple ABM with a world size of $M = 33 \times 33$ cells (the default world size in NetLogo), b) a large variant of the simple ABM with $M = 99 \times 99$ cells, and c) the enhanced ABM with $M = 99 \times 99$ cells. These environment sizes fall within research findings on the numbers of news outlets [98] in North America. For each of the three ABM instances, we set up two experiments: Case I and Case II, with a difference in initialization values for $E$ (zero for I, nonzero for II). These cases along with the full lists of initialization and configuration parameters are included in the *Github* repository [69].

Figs 3 and 4 show two examples of matching and non-matching model outcomes, respectively, as variations over 200 *ticks*, for the *SEIRS* model using 1000 runs of the simple ABM (small variant). In the matching case, the means of the ABM-runs closely follow the outcomes of the EBM (equivalence measures for all six models are in Appendix IV). Even though the ABM was initialized in the same manner at every run, the deviations of models outcomes show that infodemic dynamics are sensitive to small individual variations happening at local agent level (caused by the probabilistic conditions), whereas the EBM applies the same probabilities of state transitions at group level, i.e., mean.

Table 2 shows cumulative results (means and standard deviations) for parameter variations: $\beta, \gamma, \sigma, \xi \in [0.1; 1]$ with a step of 0.1 over 2000 *ticks*, resulting in 10 experiments for *SI*, 100 *SIS* and *SIR* each, 1000 *SIRS* and *SEIR* each, 10000

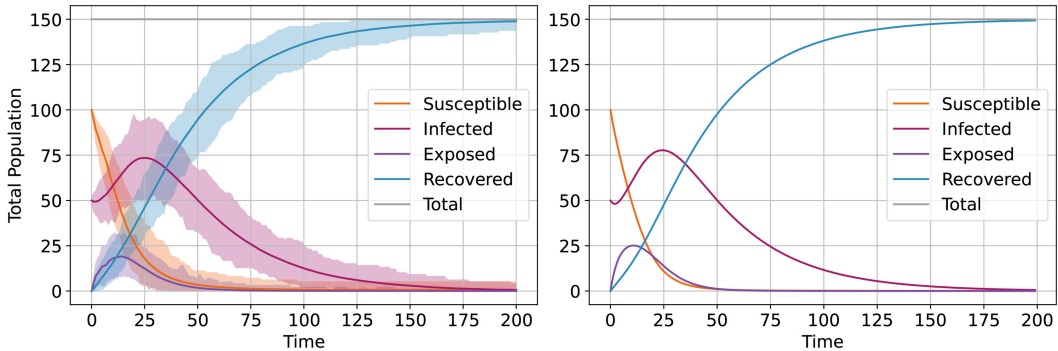

**Fig 3**. **Example of matching outcomes for the *SEIRS* model, ABM (left) vs. EBM (right),** $\beta = 0.2$, $\gamma = 0.03$, $\sigma = 0.155$, $\xi = 0.001$ (means and deviations for 1000 runs of the ABM).

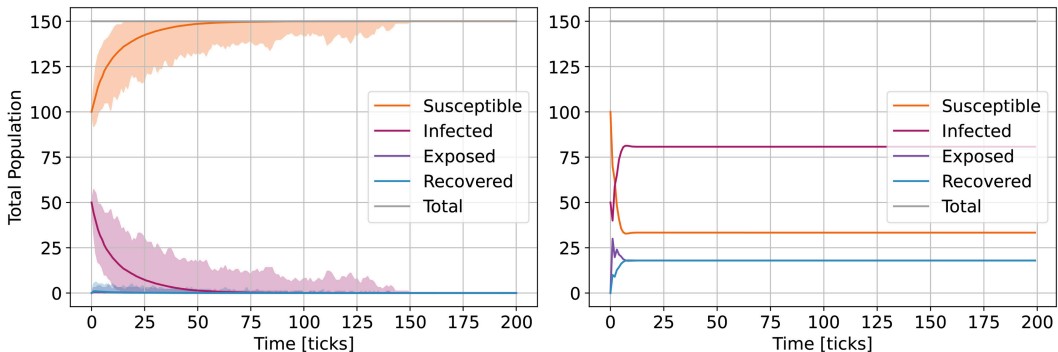

**Fig 4**. **Example of non-matching outcomes for the *SEIRS* model, ABM (left) vs. EBM (right),** $\beta = 0.9$, $\gamma = 0.2$, $\sigma = 0.9$, $\xi = 0.9$ (means and deviations for 1000 runs of the ABM).

*SEIRS*. Results are consistent for model size in the simple ABM case. Some models fare better (e.g., *SIR*) than others (e.g., *SIS*) on average, but even so, none of the EBMs reproduce exactly the ABM response across the entire parameter range. The enhanced ABM shows similar results, except for *SIR*, which is considerably less well matched.

Fig 5 shows the intermodel equivalence outcomes NRMSE and $\rho$ for models *SIS* and *SI*, obtained using parameter variations $\beta, \gamma \in [0.1; 1]$ with a step of 0.05 over 10 ABM runs each and 2000 *ticks*. The results for *S* and *I* are similar due to the mirror-effect in these two outcomes, which is expected. While there are parameter combinations and intervals for which results match, this is not consistent over their entire range. The heatmaps for the *SIS* model show combinations for which the ABM and the EBM are profoundly distinct in dynamics: an infodemic with high infection rate $\beta$ and mid-range recovery rate $\gamma$ will cause the ABM and EBM models to behave differently. The addition of the online social media network in the enhanced ABM visibly shifts the matching/non-matching coverages.

Fig 6 shows the effect of homophily (conditional infection) on the enhanced ABM vs. simple ABM outcomes for groups of different sizes $N_{G1} = 80\%$, $N_{G2} = 20\%$ (total $N = 1350$ agents), over 200 *ticks* and 1000 runs (means and deviations). While the exposed outcome seems similar enough, the others (susceptible, infected, recovered) show considerable change in infodemic dynamics. Group sizes do not show an effect (see Appendix V), but this observation is enough to raise questions regarding the many social group dynamics of the real world.

**Table 2. Cumulative results for intermodel equivalence across parameter variations:** $\beta, \gamma, \sigma, \xi \in [0.1; 1]$ **with a step of 0.1, resulting in 10 experiments for** *SI*, **100** *SIS* **and** *SIR*, **1000** *SIRS* **and** *SEIR*, **10000** *SEIRS*. Case II configuration. Notations: NRMSE normalized root mean of square error, $\rho$ Pearson's correlation coefficient, SD standard deviation, ABM agent-based model, EBM equation-based model, S susceptible, E exposed, I infected, R recovered.

| Model | NRMSE | | | | $\rho$ | | | |
|---|---|---|---|---|---|---|---|---|
| | S | E | I | R | S | E | I | R |
| Simple ABM (small variant) vs. EBM (N = 150 agents); mean (SD) | | | | | | | | |
| SI | 0.03 (0.01) | - | 0.03 (0.01) | - | 0.74 (0.16) | - | 0.74 (0.16) | - |
| SIS | 0.18 (0.24) | - | 0.18 (0.24) | - | 0.21 (0.75) | - | 0.21 (0.75) | - |
| SIR | 0.25 (0.18) | - | 0.01 (0.01) | 0.25 (0.18) | 0.68 (0.36) | - | 0.84 (0.10) | 0.90 (0.07) |
| SIRS | 0.19 (0.24) | - | 0.12 (0.18) | 0.07 (0.09) | 0.30 (0.67) | - | 0.45 (0.63) | 0.32 (0.50) |
| SEIR | 0.25 (0.15) | 0.01 (0.00) | 0.01 (0.00) | 0.25 (0.15) | 0.75 (0.35) | 0.83 (0.12) | 0.91 (0.06) | 0.94 (0.04) |
| SEIRS | 0.21 (0.26) | 0.06 (0.08) | 0.09 (0.14) | 0.06 (0.08) | 0.47 (0.61) | 0.65 (0.42) | 0.59 (0.56) | 0.42 (0.47) |
| Simple ABM (large variant) vs. EBM (N = 1350 agents); mean (SD) | | | | | | | | |
| SI | 0.04 (0.01) | - | 0.04 (0.01) | - | 0.70 (0.05) | - | 0.70 (0.05) | - |
| SIS | 0.18 (0.24) | - | 0.18 (0.24) | - | 0.14 (0.79) | - | 0.14 (0.79) | - |
| SIR | 0.30 (0.20) | - | 0.01 (0.01) | 0.30 (0.20) | 0.53 (0.39) | - | 0.82 (0.09) | 0.89 (0.07) |
| SIRS | 0.19 (0.24) | - | 0.12 (0.18) | 0.07 (0.09) | 0.24 (0.73) | - | 0.39 (0.69) | 0.28 (0.58) |
| SEIR | 0.31 (0.17) | 0.01 (0.00) | 0.01 (0.01) | 0.31 (0.17) | 0.58 (0.40) | 0.81 (0.11) | 0.90 (0.06) | 0.93 (0.04) |
| SEIRS | 0.22 (0.27) | 0.06 (0.09) | 0.10 (0.15) | 0.06 (0.09) | 0.41 (0.68) | 0.65 (0.45) | 0.55 (0.62) | 0.39 (0.54) |
| Enhanced ABM vs. EBM (N = 1350 agents); mean (SD) | | | | | | | | |
| SI | 0.03 (0.01) | - | 0.03 (0.01) | - | 0.75 (0.13) | - | 0.75 (0.13) | - |
| SIS | 0.11 (0.14) | - | 0.11 (0.14) | - | 0.35 (0.55) | - | 0.35 (0.55) | - |
| SIR | 0.20 (0.15) | - | 0.14 (0.23) | 0.26 (0.18) | 0.32 (0.29) | - | 0.36 (0.61) | 0.62 (0.29) |
| SIRS | 0.12 (0.16) | - | 0.09 (0.12) | 0.06 (0.07) | 0.37 (0.51) | - | 0.46 (0.48) | 0.36 (0.43) |
| SEIR | 0.19 (0.14) | 0.01 (0.00) | 0.17 (0.27) | 0.28 (0.21) | 0.55 (0.24) | 0.81 (0.11) | 0.38 (0.69) | 0.69 (0.31) |
| SEIRS | 0.14 (0.18) | 0.06 (0.08) | 0.11 (0.15) | 0.05 (0.06) | 0.48 (0.49) | 0.64 (0.38) | 0.54 (0.50) | 0.43 (0.44) |

## Fitting to real world data: Results

First, we applied the fitting procedure to the enhanced ABM. After 700 iterations, the best result has $\text{NRMSE}|_{S_{ABM}, S_{RW}} = 0.055$ and $\rho|_{S_{ABM}, S_{RW}} = 0.872$. The model parameter are: $\beta = 0.633$, $\gamma = 0.073$, $I_R = 0.41$, $P_M = 10.35$, $T_F = 19$, $L_M^S = 28$, $L_M^I = 4$, $D_P^S = 23$, $D_P^I = 25$, $L_P^S = 96$, $L_P^I = 22$. Second, we applied the fitting procedure to the EBM. After 700 iterations, the best result has $\text{NRMSE}|_{S_{EBM}, S_{RW}} = 0.086$ and $\rho|_{S_{EBM}, S_{RW}} = 0.714$. The model parameter are: $\beta = 338 \cdot 10^{-6}$, $\gamma = 672 \cdot 10^{-6}$, $T_F = 30$. Fig 7 shows the results of these two fitting procedures against the real world data. While the EBM outputs seem to follow an overall trend of decrease or increase, they do not capture the dynamics of people changing their opinions over time. In this case the enhanced ABM manages to show how the proportions of the population invert their ratio at weeks 18-22. This period corresponds to October 2020 when Romania faced a considerable increase in cases.

Further, we perform a comparative test with the parameters resulting from the enhanced ABM fitting procedure (where applicable). Fig 8 shows the responses of the three models (enhanced and simple ABM, EBM) against real world data. Table 3 presents the two outcomes ($\rho$, NRMSE), with the best fit for the enhanced ABM, which shows that even when the infection and recovery rates are determined to match the dynamics of real world data, the EBM response does not manage to illustrate these transient fluctuations. The simple ABM response comes closer, but the fit is still poor, highlighting the importance of the online communication network as a misinformation facilitator.

## Discussion

In this study we investigated EBMs and ABMs for the spread of mis- and disinformation. For this, we designed two models: a) a simple ABM as a direct translation of *SEIRS*-type EBMs' underlining logic, and b) an enhanced ABM to reflect

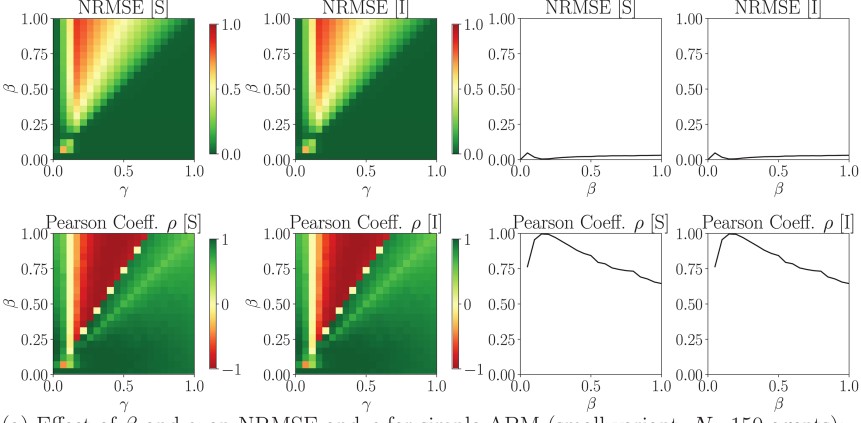

(a) Effect of $\beta$ and $\gamma$ on NRMSE and $\rho$ for simple ABM (small variant, $N$=150 agents): heatmaps for SIS (left) and graphs for SI (right)

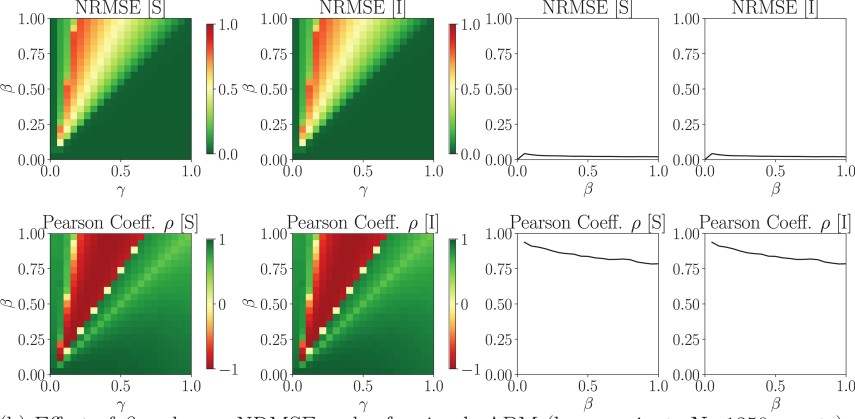

(b) Effect of $\beta$ and $\gamma$ on NRMSE and $\rho$ for simple ABM (large variant, $N$=1350 agents): heatmaps for SIS (left) and graphs for SI (right)

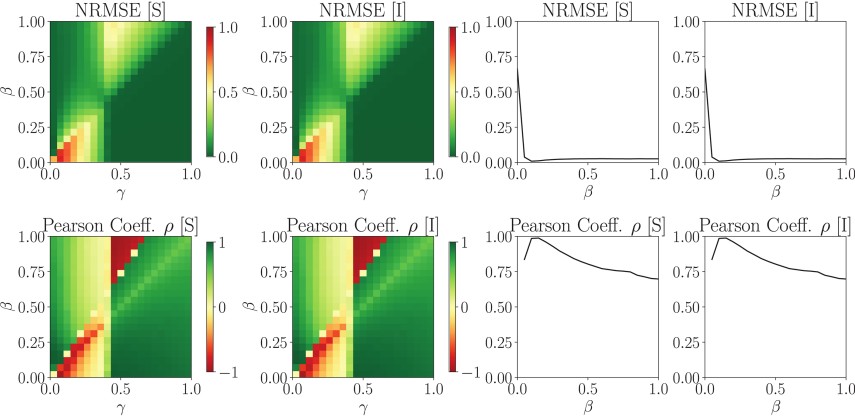

(c) Effect of $\beta$ and $\gamma$ on NRMSE and $\rho$ for enhanced ABM (large variant, $N$=1350 agents): heatmaps for SIS (left) and graphs for SI (right)

**Fig 5**. **Intermodel equivalence outcomes for the *SIS* and *SI* models (result exemplification), where NRMSE is the normalized root mean of square error, $\rho$ is Pearson's correlation coefficient, $\beta$ is the infection rate, and $\gamma$ is the recovery rate.** Case I configuration.

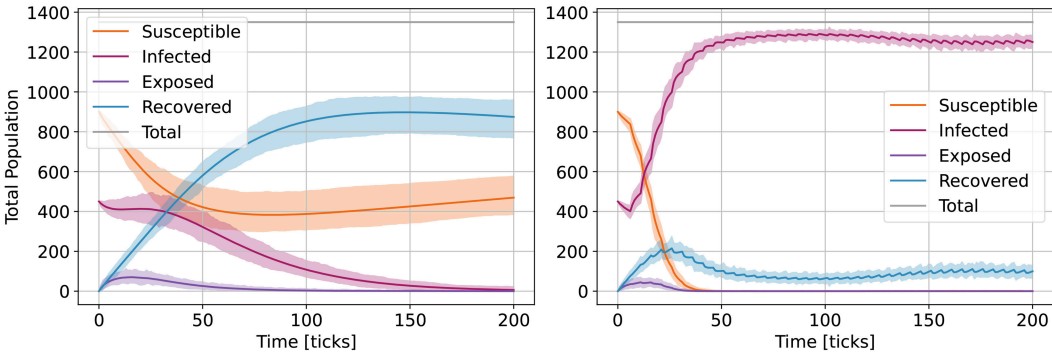

**Fig 6**. Effect of homophily on the ABM model outcomes with $N_{G1} = 80\%$, $N_{G2} = 20\%$ (total $N = 1350$ agents): simple model large variant (left) and enhanced model (right) for 1000 runs (means and deviation).

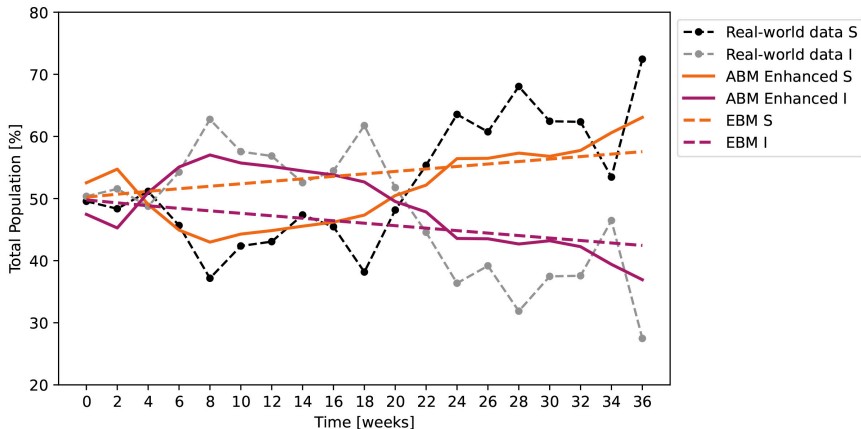

**Fig 7**. Results of the SIS model fitting against the real world data for the enhanced ABM ($\beta = 0.633$, $\gamma = 0.073$) averaged over 100 runs (variance 5498.17) and for the EBM ($\beta = 338 \cdot 10^{-6}$, $\gamma = 672 \cdot 10^{-6}$).

communication and online media networks, as well as homophily. Then, we performed an intermodel equivalence analysis. Results show that EBMs and ABMs display both matching and non-matching outcomes, depending on parameter ranges. Our hypothesis that microscopic models (e.g., ABMs) are necessary to capture the elements of the human psycho-social context is confirmed.

## ABM and EBM discrepancies

The critical difference between these types of models is that EBMs assume individual homogeneity, whereas the ABM structure allows for more complex reasoning, such as homophily, multiple pieces/types of information, believing views opposite their own group, etc. While there are attempts to rework EBMs for various epidemics [51,99], in some cases combining epidemics and infodemics in one state machine [100], these models have the same issue: losing individuality.

The main reason for the mismatch of these types of models is that the effects observed at population level (e.g., infection rate) are not observed at individual level (e.g., probability of infection) because humans, as psycho-social networks, behave like a complex system in which the infodemic is an emergent process, and to paraphrase Kevin Kelly [1], even made of the same building blocks, we would not find the beehive in the bee. The top-down approach of equation-based modeling transforms a population into one black-box mass, whereas the bottom-up design of complex systems allows for

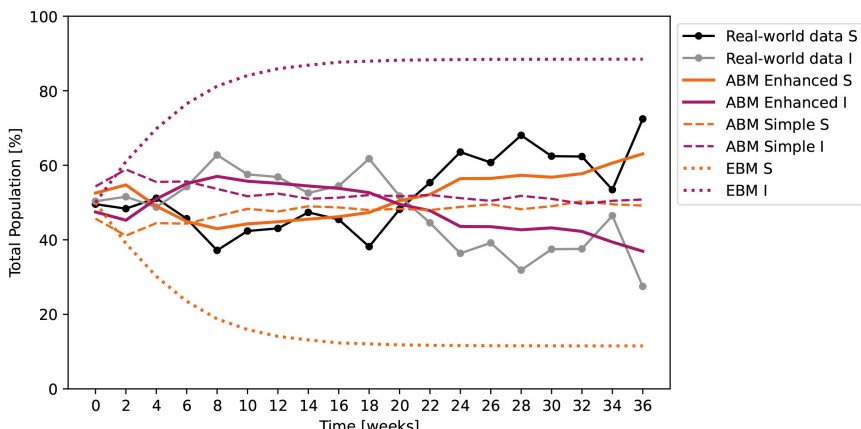

**Fig 8**. **Enhanced ABM outputs fitted against real world data, compared with the simple ABM and the EBM outputs obtained using the same parameters (where applicable).** $\beta = 0.633$, $\gamma = 0.073$. Both ABM outputs are averaged over 100 runs and have a variance of 5498.17 for the enhanced and 1577.25 for the simple versions.

**Table 3**. **Outcome measures (Pearson correlation coefficient $\rho$ and normalized root mean of squared error NRMSE) comparing the real-world data with the outputs of the enhanced ABM, simple ABM and EBM, simulated with the parameters $\beta = 0.633$, $\gamma = 0.073$, resulted from the enhanced ABM fitting procedure.** Notations: ABM agent-based model, EBM equation-based model.

|  | **ABM Enhanced** | **ABM Simple** | **EBM** |
|---|---|---|---|
| $\rho$ | **0.872** | 0.391 | -0.067 |
| NRMSE | **0.055** | 0.103 | 0.418 |

these diffusion processes to emerge naturally, from interactions between individuals. Strong emergence is the "magic" that can explain multi-layered human interactions at large scale [101]. This is why, when the conditions allow it, the ABM approach would be preferable to analyze, for instance, the effects of individual vs. mass-level interventions of information manipulation (e.g., propaganda).

The *SEIRS*-type EBMs we analyzed here are not necessarily entirely wrong, but they are based on assumptions that do not consider the whole nature of information diffusion. Epidemics driven by biological viruses assume some level of physical contact, whereas information spreads through entirely different media and mechanics. Epidemic research has modified these basic models to better represent various diseases, and the same should happen for infodemics. As they are now, the *SEIRS* EBM models are missing the communication and online media networks, as well as psycho-social attachments and beliefs, meaning that each person would have different infection rates based on information coming from different sources.

## Implications of the results

So, then, what? Models cannot be perfect imitations of the real world [102], but their level of wrongness matters when humans are involved. Infodemic and epidemic models are useful for testing in a safe environment (simulation) various interventions before deployment or adoption, and it is critical that the model predictions are as accurate as possible.

Concerning the corrective interventions on mis- and disinformation, EBMs do not currently offer ways to simulate or evaluate these mechanisms, as they only show the outcomes of a population (or subset). Efficiency estimations of local interventions require ABMs. Often, interventions to resist misinformation must overcome various cognitive and socio-affective barriers. The most common types of correction are individual. Fact-based corrections directly address inaccuracies and provide accurate information [103]. Broader protection against different types of misleading tactics is offered

by addressing the logical fallacies common in disinformation [104] or by challenging the plausibility of the misinformation or the credibility of its source. Multiple approaches can be combined into a single correction [105]; however they all have to be applied locally. In fact, effective regulatory actions must be implemented at individual level. Even though some corrections can propagate through specific social groups, there is always an element of internally-generated intent if coercive action at the population level is to be avoided. This is what in systems science is known as decentralized control, as opposed to centralized, in which an authority body applies interventions at societal level.

As it currently stands, EBMs can still be useful. Because they provide mean trends within a population, EBMs can supplement predictive ABM simulations and act as baselines for identifying unusual, unexpected, or outlier behaviors. However, we suggest that EBMs should be revised when it comes to misinformation diffusion, so that the parameters of the wide-scale infodemic would reflect the emergence from local behaviors. As the case study on vaccine acceptance shows, the enhanced ABM fits the real world data better than the simple version. This suggests that adding elements to a model can be helpful and sometimes even necessary to correctly interpret the behavior of the modeled system, especially when it is generated from non-trivial interactions, such as information diffusion in an infodemic. Future studies should consider the limitations and advantages of both ABMs and EBMs, and keep in mind that their outcomes are not always equivalent.

## Effect of the number of parameters

The enhanced ABM includes a larger number of parameters compared to the EBM, due to the increase in structural complexity required to simulate individual-level mechanisms, which is absent when a system is studied in mean-field. Parameters such as energy dynamics, group membership, and environmental information flows represent psycho-social and communication-related processes non-modelable in equation-based frameworks. To guarantee the same parameter space would mean to lose adherence to real-world dynamics; thus, the increased dimensionality is not a modeling convenience but a representational necessity.

Moreover, we believe that the superior performance of the ABM in fitting real-world data does not merely arise from parameter abundance, but from a capacity to replicate emergent dynamics enabled by its structure. For instance, the enhanced ABM reproduces non-monotonic behaviors observed in vaccine acceptance data, including an inversion in public sentiment corresponding to a real-world event. These nonlinear transitions are a product of local agent interactions which can not be encoded in the EBM's aggregate-level formulation while maintaining a correspondence with the modeled system. In this sense, the ABM does not merely fit better—it explains better.

Also, while in classical fitting problems the fitness typically increases with the number of parameters used to tune the model's behavior, this relationship does not always hold in causal models in general, and in ABMs specifically. If the underlying structure is not well designed—meaning it fails to adequately represent the cause-effect relationships within the system in some stylized functional form—then adding new parameters does not necessarily improve the fit; in fact, it can even reduce it.

Finally, the fitting to empirical data was conducted via a multi-objective optimization procedure that penalizes poor generalization through metrics such as $\rho$ and NRMSE. With regard to intermodel equivalence assessment, the combination of these metrics provide a summary perspective on outcome similarity. We chose them for their capacity to describe the timewise longitudinal relationships between discrete signals through single-scalar quantities that are easily interpretable through interoperable terminology across fields of science (modeling, statistics, sociology, engineering, etc.). These metrics do present limitations. Pearson's correlation coefficient $\rho$ is famously sensitive to the smallest differences, which we mitigate with the rougher NRMSE. Nonlinear correlations will not be captured by $\rho$ and NRMSE—but because the two model types would represent the same phenomenon/process, we should not find noncausal nonlinearities between their outcomes, which reinforces our conclusion that infodemic EBMs must be redefined.

## Conclusions and future developments

In this study, we evaluated the intermodel equivalence for infodemics. We designed agent-based models (ABMs) and compared their outcomes with classical equation-based models (EBMs) inspired by viral epidemics. We found low equivalence over the entire key parameter range, although the outcomes were similar for specific values. We also found that ABMs can capture the dynamics of real world data better than EBMs. We surmise that ABMs and EBMs serve different purposes with widely different structures (one is microscopic, the other macroscopic) and the decision of choosing one over the other should be informed, with awareness to their limitations and the fact that they are not interchangeable. Moreover, EBMs for infodemics should be revised from their counterparts modeling biological viral spreads.

Future developments include building an interaction topology between agents on a network and subsequently assessing how the structure of interactions (i.e., the features of the network) and its dynamics (creating or destroying links between agents) affect the results of the equation- and agent-based diffusion models. Moreover, we will implement and investigate time- or location-variant parameters (e.g., variable infection rate), introduce vital dynamics to both types of models and mitigation mechanisms, as well as different pieces of mis- and disinformation spreading concurrently through the agent population.

## Supporting information

**Appendix I. Equation-based SEIRS-class models.** List of the equation-based models and their parameters, together with the state transition representations.
(PDF)

**Appendix II. Implementation.** The flow diagram for model scheduling and the interfaces of the two ABMs.
(PDF)

**Appendix III. Algorithms.** Definitions and algorithms specific to the implementation fo the two ABMs.
(PDF)

**Appendix IV. Examples of matching outcomes.** Parameters and visualization of matching outcomes examples for the six SEIRS-class models.
(PDF)

**Appendix V. Selected results (exemplification).** Examples for the effect of homophily.
(PDF)

**Repository. Understanding the Mechanisms of Infodemics: Equation-Based vs Agent-Based Models.** Models, experiment configurations, and heatmap results: https://github.com/cristian-berceanu/understanding_the_mechanisms_of_infodemics

**Video. Simulation demonstration.** A brief video demonstration of the two ABMs during simulation.
(MP4)

## Author contributions

**Conceptualization:** Cristian Berceanu, Francesco Bertolotti, Nadia Arshad, Monica Patrascu.

**Data curation:** Cristian Berceanu, Monica Patrascu.

**Formal analysis:** Cristian Berceanu, Francesco Bertolotti, Monica Patrascu.

**Investigation:** Cristian Berceanu, Francesco Bertolotti, Nadia Arshad, Monica Patrascu.

**Methodology:** Cristian Berceanu, Francesco Bertolotti, Nadia Arshad, Monica Patrascu.

**Resources:** Cristian Berceanu, Monica Patrascu.

**Software:** Cristian Berceanu, Francesco Bertolotti, Monica Patrascu.

**Supervision:** Monica Patrascu.

**Validation:** Cristian Berceanu, Francesco Bertolotti, Nadia Arshad, Monica Patrascu.

**Visualization:** Cristian Berceanu, Francesco Bertolotti, Nadia Arshad, Monica Patrascu.

**Writing – original draft:** Cristian Berceanu, Francesco Bertolotti, Nadia Arshad, Monica Patrascu.

**Writing – review & editing:** Cristian Berceanu, Francesco Bertolotti, Nadia Arshad, Monica Patrascu.

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
