## [Decision Letter · Decision Letter 0]

22 Apr 2025

PONE-D-24-58654Understanding the mechanisms of infodemics: equation-based vs. agent-based modelsPLOS ONE

Dear Dr. Patrascu,

Thank you for submitting your manuscript to PLOS ONE. After careful consideration, we feel that it has merit but does not fully meet PLOS ONE’s publication criteria as it currently stands. Therefore, we invite you to submit a revised version of the manuscript that addresses the points raised during the review process.

We look forward to receiving your revised manuscript.

Kind regards,

William Waites

Academic Editor

PLOS ONE

Journal Requirements:

3. We note that your Data Availability Statement is currently as follows: All relevant data are within the manuscript and in Supporting Information files.

Additional Editor Comments:

Dear Dr Patrascu,

Thank you for this article and my apologies for the very long time it has taken to handle it. This is entirely my fault. The reviewers have brought up some important criticisms of the manuscript so I am returning it to you for revision. In addition to the comments of the reviewers, I wonder if you have considered the related work in epidemics on networks, for example the excellent book by Miller and Kiss. The reason I mention this is that there is a spectrum from well-mixed ODE systems through to individually structured populations which can be represented on a network. There are well-known results about how the dynamics change with the introduction of this kind of population structure (essentially, the epidemic will proceed more slowly and there can be depletion of susceptible individuals and in some cases patterns not unlike what you find in the continuous case with reaction-diffusion systems can arise). The cell-based ABM formulation is a kind of middle ground where interactions are locally well-mixed but with different localities (cells, in the ABM) but with transport between the cells. It would be interesting to see the work better situated in this landscape of modelling techniques.

With best wishes,

William Waites

Reviewers' comments:

Reviewer's Responses to Questions

**Comments to the Author**

1. Is the manuscript technically sound, and do the data support the conclusions?

Reviewer #1: Partly

Reviewer #2: Partly

2. Has the statistical analysis been performed appropriately and rigorously? 

Reviewer #1: I Don't Know

Reviewer #2: No

3. Have the authors made all data underlying the findings in their manuscript fully available?

Reviewer #1: Yes

Reviewer #2: No

4. Is the manuscript presented in an intelligible fashion and written in standard English?

Reviewer #1: Yes

Reviewer #2: No

5. Review Comments to the Author

Reviewer #1: The background is generally well-written but could benefit from improved structure. Divide the section into subsections for better organization (e.g., "Diffusion Models Overview," "Epidemic Models," "Information Spread Models").

Can use bullet points or tables to summarize and compare Equation-Based Models (EBMs) and Agent-Based Models (ABMs) for clarity.

The motivations and aims of the paper are clear. However better structuring (e.g., bullet points for the key objectives of EBM and ABM comparisons) would enhance clarity and focus. The connection between the discussion on psychological perspectives and the modeling process is unclear.

Clearly state how the points raised in the psychological perspectives section are incorporated into the model design. For instance, explicitly link psychological and psycho-social components to specific parameters, mechanisms, or agent behaviors.

The discussion of (any) stochasticity in the ABM model is insufficient. Clarify how stochasticity is implemented (e.g., specific features of the NetLogo platform). Provide examples of how randomness may affect the results.

Some technical aspects of the implementation (e.g., the use of energy) may overwhelm readers and can be moved into appendix.

Comparative Analysis (ABM vs. EBM). The section lacks context (why is embedded there), and the motivations for performing the comparative analysis are unclear.

Explain why this comparison is important and how it informs the rest of the paper.

Clearly state how the findings from this section contribute to understanding the differences and complementarities between ABMs and EBMs.

On Page 10, clarify what S_RW and I_RW represent.

In Equation (4), explain the rationale for using the "min over k."

The term “preferential attachment” is used differently from its standard meaning in network science. Consider using a different term to avoid confusion. Clarify whether this mechanism is deterministic in your context.

To my understanding, the ABM has more parameters than the EBM, which could (inherently) make it easier to achieve a better fit to real data. Discuss this important potential limitation of the paper more explicitly.

On the same point, explore methods for a "fair" comparison between the two approaches, such as focusing on a small subset of key parameters in the ABM to identify those most critical for matching real data.

Reviewer #2: The paper “Understanding the mechanisms of infodemics: equation-based vs. agent-based models” presents a comparison of the two types of models (equation- vs agent-based) applied to the scenario of mis- and disinformation spreading, on the established analogy of epidemiologic modelling. The models are compared to each other on a wide range of variants (SI, SIS, etc.) and comprehensive parameter range, and also fitted to a real-world desinformation spread scenario. The conclusions present both the comparison of the models run, as well as the evaluation of the real-world data fitting. The code is well-written and organised and easy to check.

On the whole the objective of the paper is valid and worth the publication, and the authors completed much of the necessary groundwork for the research. However, currently there are several points that need further work and clarification.

-- The linguistic level of the presentation on the whole needs attention. Sentences like the below hinder the understanding and when revisited and improved, can add to the overall value of the paper.

“Appendix II presents in the details the 359 models scheduling and their interfaces.”

-- The framework in which the models are compared needs re-evaluation, as well as a clearer presentation.

* The way the two models are compared and evaluated should be clarified. The paper uses two metrics (Pearson’s correlation and normalized root mean square error), but neither of them are well motivated in the paper.

* Pearson’s correlation coefficient only measures linear dependency, and it can have the value zero for dependent variables (if the dependence is not linear). It is not clear why we expect linear dependency: please, either clarify this or perhaps use Brownian (distance) covariance.

* Normalized root mean of square error, as presented in the paper, is not symmetric, as the denominator only depends on one side. It is a strange property for a distance metric. Again, either clarify this choice or use a symmetric normalisation.

* Since there are two distance metrics (Pearson and NRMSE) I would expect a discussion of their comparison. This should include their theoretical properties, as well as how they relate to the fitting process, and perhaps a discussion of their differences across the outcomes.

-- It is not clear what metrics the fitting process relies on. I think it needs some clarification.

-- The paper’s conclusion about the preference of ABMs is too optimistic in my opinion. Since the ABMs have more parameters, they are expected to fit real-world data better, which means that this in itself does not evidence the explanatory power of the model. I think an open discussion of the modest conclusions such experiments can support would not reduce the value of the research but rather strengthen it.

-- The result csv files of the model comparison have been uploaded to git, but I was unable to find the actual piece of code that generates these files. More importantly, I did not find the data on which the parameters were fitted. Compiling the whole case study into a one-liner that makes the flow of the work easy to follow would be very useful to enable the transparent presentation of the findings and would help future researchers rely on the results. Making everything executable by a docker file or something similar would be very useful for easy reproducability.

With the above changes the paper will be a solid contribution to the journal.

6. PLOS authors have the option to publish the peer review history of their article (what does this mean?). If published, this will include your full peer review and any attached files.

Reviewer #1: No

Reviewer #2: No

---

## [Author Response · Author response to Decision Letter 1]

11 Jul 2025

Response to Reviewers (formatted letter uploaded as attachment)

June 2025

Dear editors and reviewers,

Thank you for the opportunity to revise our manuscript, and for the thorough comments. We have revised the paper in light of these useful suggestions. Our responses and changes to the manuscript can be found below (point-by-point for each item) and are as well highlighted in the uploaded revised manuscript.

1 Editor

In addition to the comments of the reviewers, I wonder if you have considered the related work in epidemics on networks, for example the excellent book by Miller and Kiss. The reason I mention this is that there is a spectrum from well-mixed ODE systems through to individually structured populations which can be represented on a network.

Thank you for your comment. We are familiar with the book, which we consider an excellent reference in the field. The aim of our work was to explore whether it is possible to represent the same diffusion phenomenon using models with fundamentally different structures but identical parameters. A possible direction (that we have investigated) would have been to include a networkbased model as an intermediate step between a fully mixed ODE system and an agent-based approach. This still remains in our field of interest for further research, and at page 4 there is a review of these elements.

There are well-known results about how the dynamics change with the introduction of this kind of population structure (essentially, the epidemic will proceed more slowly and there can be depletion of susceptible individuals and in some cases patterns not unlike what you find in the continuous case with reaction-diffusion systems can arise).The cell-based ABM formulation is a kind of middle ground where interactions are locally well-mixed but with different localities (cells, in the ABM) but with transport between the cells. It would be interesting to see the work better situated in this landscape of modelling techniques.

Thank you for the suggestion. We have revised the text and included additional references to clarify the context, as you advised. In particular, we now explicitly mention key works in the field, such as the seminal paper by Newman (2002), the review by Keeling and Eames (2005), and the comprehensive overview by Duan et al. (2015). This should make our approach and its position within the broader landscape of modeling techniques more transparent.

2 Reviewer 1

The background is generally well-written but could benefit from improved structure. Divide the section into subsections for better organization (e.g., ”Diffusion Models Overview,” ”Epidemic Models,” ”Information Spread Models”).

Thank you for your positive opinion and constructive feedback. We appreciate and had implemented your suggestion to better divide the background and the rest of the paper, for a greater readability.

Can use bullet points or tables to summarize and compare Equation-Based Models (EBMs) and Agent-Based Models (ABMs) for clarity.

Thank you for the suggestion, we appreciate the input. At the end of the models description section, there is now a summary table outlining the differences between EBM, simple ABM, and enhanced ABM. (Table 1)

The motivations and aims of the paper are clear. However better structuring (e.g., bullet points for the key objectives of EBM and ABM comparisons) would enhance clarity and focus.

Thank you for pointing this out. We restructured the aims and listed the key objectives under their two categories. (lines 48-64) (view Page 3)

The connection between the discussion on psychological perspectives and the modeling process is unclear. Clearly state how the points raised in the psychological perspectives section are incorporated into the model design. For instance, explicitly link psychological and psycho-social components to specific parameters, mechanisms, or agent behaviors.

Thank you for this comment. We added an explanatory note at the end of the psychological perspectives section. We hope this clarifies how we made our design choices, as we wanted to avoid repeating some of the technical methodological choices we describe in the next sections, which follow directly from the psych analysis. (view page 6 in section ”Preliminary Analysis: a Psychological Perspective on the Infodemic Infection Mechanisms” )

The discussion of (any) stochasticity in the ABM model is insufficient. Clarify how stochasticity is implemented (e.g., specific features of the NetLogo platform). Provide examples of how randomness may affect the results.

Thank you for catching this. We revised the section describing the intermodel comparison and clarified how we handle stochasticity across experiments (combined with the comment on fairness of the comparison). We provide examples of how randomness can affect results in the Results section, figures 3 and 4, with their associated explanations in text, where we show different variations around the mean for multiple ABM runs. Further examples of this are included in Appendices IV and V (page 15 in Result: Evaluation of intermodel equivalence ).

Some technical aspects of the implementation (e.g., the use of energy) may overwhelm readers and can be moved into appendix.

Thank you for this comment. As energy is a state variable that we chose as a result of the psychological perspective analysis, we believe it should be described together with the other state variables. We have now linked this variable with the explanation on how the psych perspective has informed our design choices. (view Appendix II)

Comparative Analysis (ABM vs. EBM). The section lacks context (why is embedded there), and the motivations for performing the comparative analysis are unclear. Explain why this comparison is important and how it informs the rest of the paper. Clearly state how the findings from this section contribute to understanding the differences and complementarities between ABMs and EBMs.

We agree that the rationale for directly comparing ABMs and EBMs should be stated more clearly. We have revised the introduction of the section to clarify that this comparison is essential because EBMs are the traditional standard for modeling diffusion phenomena (especially when the number of entities in the system exceeds a threshold and make it hard to simulate on the computational side), but their underlying assumptions—such as population homogeneity and mean-field approximations—may overlook important individual-level mechanisms captured by ABMs. Systematically comparing the two approaches across a broad parameter range and fitting them to real-world data allows a better understand where they diverge and under what conditions each is more appropriate. So the findings from this section help clarify the specific strengths and limitations of both modeling paradigms, thereby informing the choice of method for future research on information diffusion and infodemics. Now, these elements are clearly highlighted in the section. (view page 12 in Comparative Analysis)

On Page 10, clarify what SRW and IRW represent.

We now specified these variables.

In Equation (4), explain the rationale for using the ”min over k.”

It was a typo, we meant min over πi, which is the set of parameters over which the optimization has been performed. Thank you for noticing it, now it is corrected. (page 13 now corrected)

The term “preferential attachment” is used differently from its standard meaning in network science. Consider using a different term to avoid confusion. Clarify whether this mechanism is deterministic in your context.

Thank you for this point, we changed it to homophily, to illustrate a preference toward members of the same group. We clarified how the group sizes and assignments are specified in the simulations (combined with the comment regarding stochasticity).

To my understanding, the ABM has more parameters than the EBM, which could (inherently) make it easier to achieve a better fit to real data. Discuss this important potential limitation of the paper more explicitly.

Thank you for pointing out this important limitation. We have now added a dedicated section in the Discussion to address it . In brief, we clarify that the higher number of parameters in the enhanced ABM reflects its structural complexity, necessary to capture individual-level mechanisms and emergent dynamics, and that it is this better structural model adherence to the modelled system that explains the improved fit. (page 20 in Discussion)

On the same point, explore methods for a ”fair” comparison between the two approaches, such as focusing on a small subset of key parameters in the ABM to identify those most critical for matching real data.

In this study we implemented the simple ABM as the fair comparison, as other functions associated with the complexity of human information exchanges are not included. To the same effect, the inter-model comparative analysis is reduced to the key parameters of the EBMs (e.g., infection rate, etc.), while the other ABM parameters are fixed - we now clarified this in the methodology. We also fixed the initialization of the ABM to reduce stochasticity, and also averaged the ABM experiments over several runs (standard practice in complex systems simulation). For the real world data, our goal was to assess the full power of each model, and so all parameters were utilized, with the same initialization of the map/seed; here, we included the time factor parameter to allow for a fair longitudinal sampling across models relative to the sampling of the real world data. The discussion can be found at page 18.

3 Reviewer 2

The paper “Understanding the mechanisms of infodemics: equation-based vs. agent-based models” presents a comparison of the two types of models (equationvs agent-based) applied to the scenario of mis- and disinformation spreading, on the established analogy of epidemiologic modelling. The models are compared to each other on a wide range of variants (SI, SIS, etc.) and comprehensive parameter range, and also fitted to a real-world desinformation spread scenario. The conclusions present both the comparison of the models run, as well as the evaluation of the real-world data fitting. The code is well-written and organised and easy to check.

Thank you for your review. We truly appreciate the time and effort you dedicated to understanding the structure and aims of the paper, as well as you appreciation for the paper organization and the quality of the code.

On the whole the objective of the paper is valid and worth the publication, and the authors completed much of the necessary groundwork for the research. However, currently there are several points that need further work and clarification.

Thank you for acknowledging the relevance of our work and pointing out elements that could be improved to make it a better scientific output.

– The linguistic level of the presentation on the whole needs attention. Sen-tences like the below hinder the understanding and when revisited and improved, can add to the overall value of the paper.“Appendix II presents in the details the 359 models scheduling and their interfaces.”

Thank you for pointing this out. We performed a language check throughout the paper and revised awkward sentences/typos.

– The framework in which the models are compared needs re-evaluation, as wellas a clearer presentation.

Thank you for these critical comments. We revised the section describing the comparison and metrics (we answer the bullets points one by one below).

* The way the two models are compared and evaluated should be clarified. Thepaper uses two metrics (Pearson’s correlation and normalized root mean square error), but neither of them are well motivated in the paper.

Beside the elements described in the next bullet points, we added an explanation of why we chose the metrics and how we interpret them during the evaluation (pag. 13).

* Pearson’s correlation coefficient only measures linear dependency, and it canhave the value zero for dependent variables (if the dependence is not linear). It is not clear why we expect linear dependency: please, either clarify this or perhaps use Brownian (distance) covariance.

Thank you for this comment, indeed the Pearson correlation measures linear dependency. But here we do not aim to check wether the two variables are dependent, as they result from two different models. What we test is the dependence of samples over time. This technique is often used in signal processing to check signal (or time series) similarity. To make a parallel with the statistical approach, we are checking the dependence between two outcome measures collected from two different populations/groups, cross-sectionally, over many time points (q datapoints in our case). We added an explanation in the text to clarify this design (pag. 13).

* Normalized root mean of square error, as presented in the paper, is not sym-metric, as the denominator only depends on one side. It is a strange property for a distance metric. Again, either clarify this choice or use a symmetric normalisation.

Thank you for catching this. The denominator in this case refers to the minimum and maximum of either variable - for our case, they are the same because the total population is constant. We fixed the notation and clarified (pag. 12 of the manuscript).

* Since there are two distance metrics (Pearson and NRMSE) I would expect adiscussion of their comparison. This should include their theoretical properties, as well as how they relate to the fitting process, and perhaps a discussion of their differences across the outcomes.

We described the properties of the metrics under the motivation for their selection, with a clarification under the fitting to real-world data section (answered in the previous bullet points, and can be found in the text at page 12). To the Discussion section (pag. 20, at the end of the Effect of the number of parameters subsection), we added limitations and implications for their use in this work. We do not compare the metrics since we use them as a combined assessment - we clarified this in the Methods as we responded to the comments of this list.

– It is not clear what metrics the fitting process relies on. I think it needs some clarification.

We now revised and fixed the typo in Eq. 4n (pag 13), and together with the clarification regarding the metrics, we hope this is clarified. Generally, there are many metrics that can be used in model fitting to real world data; for simplicity and to avoid overwhelming the reader, we utilized the same metrics as for the inter-model comparison. To calculate the two metrics (Pearson’s coefficient and NRMSE), we frame the optimization function objectives according to the size of the real-world dataset (clarified in text), and introduce the time factor parameter to adapt their sampling times to the one of the real-world data.

– The paper’s conclusion about the preference of ABMs is too optimistic inmy opinion. Since the ABMs have more parameters, they are expected to fit real-world data better, which means that this in itself does not evidence the explanatory power of the model. I think an open discussion of the modest conclusions such experiments can support would not reduce the value of the research but rather strengthen it.

We have now included a specific section in the Discussion (pag. 20, subsection Effect of the number of parameters) where we address this issue, particularly to justify our statement. The main idea is that the improved fit results from the model’s ability to better represent the underlying real-world dynamics, rather than simply from having more parameters—which, in the case of ABMs, does not necessarily lead to better fitness outcomes.

– The result csv files of the model comparison have been uploaded to git, butI was unable to find the actual piece of code that generates these files. Compiling the whole case study into a one-liner that makes the flow of the work easy to follow would be very useful to enable the transparent presentation of the findings and would help future researchers rely on the results. Making everything executable by a docker file or something similar would be very useful for easy reprodu

---

## [Decision Letter · Decision Letter 1]

10 Sep 2025

PONE-D-24-58654R1Understanding the mechanisms of infodemics: equation-based vs. agent-based modelsPLOS ONE

Dear Dr. Patrascu,

Thank you for submitting your manuscript to PLOS ONE. After careful consideration, we feel that it has merit but does not fully meet PLOS ONE’s publication criteria as it currently stands. Therefore, we invite you to submit a revised version of the manuscript that addresses the points raised during the review process.

Reviewer 3 has suggested some minor but useful comments. Please revise your manuscript as suggested.

We look forward to receiving your revised manuscript.

Kind regards,

Siew Ann Cheong, Ph.D.

Academic Editor

PLOS ONE

Journal Requirements:

Reviewers' comments:

Reviewer's Responses to Questions

**Comments to the Author**

1. If the authors have adequately addressed your comments raised in a previous round of review and you feel that this manuscript is now acceptable for publication, you may indicate that here to bypass the “Comments to the Author” section, enter your conflict of interest statement in the “Confidential to Editor” section, and submit your "Accept" recommendation.

Reviewer #1: All comments have been addressed

Reviewer #3: (No Response)

2. Is the manuscript technically sound, and do the data support the conclusions?

Reviewer #1: Yes

Reviewer #3: Yes

3. Has the statistical analysis been performed appropriately and rigorously? 

Reviewer #1: Yes

Reviewer #3: (No Response)

4. Have the authors made all data underlying the findings in their manuscript fully available?

Reviewer #1: Yes

Reviewer #3: (No Response)

5. Is the manuscript presented in an intelligible fashion and written in standard English?

Reviewer #1: Yes

Reviewer #3: (No Response)

6. Review Comments to the Author

Reviewer #1: All comments have been addressed and the manuscript has been improved. I believe can now be published

Reviewer #3: In this paper, the authors present a comparative study of equation-based models (EBMs) and agent-based models (ABMs) to investigate the diffusion mechanisms of misinformation and disinformation, particularly in the context of infodemics during global crises such as the COVID-19 pandemic. The study is grounded in six classical SEIRS-class models and introduces both a simple ABM and an enhanced ABM that incorporates psycho-social dynamics and communication networks. The authors conduct an extensive set of simulations (11,110 experiments) and further validate their models using real-world vaccine acceptance data over 36 weeks.

The work is timely, highly relevant, and well-written. However, to be considered for publication, the authors are encouraged to address the following comments, which aim to enhance the overall quality and clarity of the paper. PLEASE SEE ATTACHED FILE

7. PLOS authors have the option to publish the peer review history of their article (what does this mean?). If published, this will include your full peer review and any attached files.

Reviewer #1: No

Reviewer #3: No

---

## [Author Response · Author response to Decision Letter 2]

27 Oct 2025

Dear editors and reviewers,

Thank you for the opportunity to revise our manuscript, and for the thorough comments. We have revised the paper in light of these useful suggestions. Our responses and changes to the manuscript can be found below (point-by-point for each item) and are as well highlighted in the uploaded revised manuscript.

Reviewer #1

All comments have been addressed and the manuscript has been improved. I believe can now be published

Answer: Thank you for taking the time to review our manuscript. Your feedback has substantially improved our work.

Reviewer #3

In this paper, the authors present a comparative study of equation-based models (EBMs) and agent-based models (ABMs) to investigate the diffusion mechanisms of misinformation and disinformation, particularly in the context of infodemics during global crises such as the COVID-19 pandemic. The study is grounded in six classical SEIRS-class models and introduces both a simple ABM and an enhanced ABM that incorporates psycho-social dynamics and communication networks. The authors conduct an extensive set of simulations (11,110 experiments) and further validate their models using real-world vaccine acceptance data over 36 weeks. The work is timely, highly relevant, and well-written. However, to be considered for publication, the authors are encouraged to address the following comments, which aim to enhance the overall quality and clarity of the paper.

Answer: Thank you for this thorough review and the appreciation of our manuscript. Your feedback has substantially improved our work.

Abstract:

Your abstract should be more explicit to encourage readers to engage with the entire paper. Please consider the following suggestions:

1- The first sentence is somewhat generic; you could make it more compelling by highlighting the impact of false information or by referencing a specific example. For example: “In an era where digital communication accelerates the global spread of false narratives, understanding how misinformation and disinformation propagate, especially during crises such as the COVID-19 pandemic, is vital to public health and policy.”

2- Make the contribution stand out more explicitly by showing what’s new here? For example: “We introduce a novel enhanced ABM that integrates psycho-social factors and communication networks, which are elements often overlooked in traditional EBM frameworks.”

3- The final sentence is informative but could be more assertive in emphasizing the implications. For example: “These findings underscore the critical role of model structure in capturing infodemic dynamics, and advocate for the use of ABMs when psycho-social influences and network interactions are central to the phenomenon.”

Answer: Thank you for these suggestions, they make the abstract more explicit. We integrated the suggestions and revised the abstract.

Methods

1. Can you include this link (https://github.com/cristi92b/infodemic_models_comparison) as a footnote with a number instead of putting it explicitly in the text? See also line 566.

Answer: According to the journal guidelines: "Footnotes are not permitted. If your manuscript contains footnotes, move the information into the main text or the reference list, depending on the content." We thus created a bibliographical entry for the repository.

2. Each equation should end with either a comma or a full stop. When presenting multiple equations, use commas after each one except the last, which should end with a full stop. The following paragraph must then begin with a capital letter. For example, Equation (1), spanning lines 1–3 of the ODE, should end with a comma; line 4 should end with a full stop, and the word “where” should begin with a capital letter: “Where.” Please review the remaining equations and apply the same formatting consistently.

5. Each equation should be preceded by a specific reference number. For instance, line 239 you should put à text like « ... as shown in Eq. (1) ».

Answer: We revised both the style of formatting equations and their referencing in text. Thank you for catching this.

3. Your methodology could be strengthened by including a brief description of the agentbased model (ABM) using the ODD (Overview, Design Concepts, and Details) Protocol proposed by Grimm et al. (https://www.jasss.org/23/2/7.html). This protocol offers a comprehensive framework for clearly communicating ABMs. It facilitates understanding and reproducibility of the model’s design and implementation, and helps researchers, practitioners, and non-experts engage with the work more effectively. Incorporating the ODD protocol will help novice readers better grasp your methodology and findings. For the Simple ABM and the Enhancing ABM, you can:

a. Overview: Describe the purpose, scope, and basic functionality of the model (…).

b. Design Concepts: Include key elements such as agents, agent behavior, environment, and the info spread process (…)

c. Details: Provide information on parameters, initialization, and the simulation process (…)

Answer: Thank you for this suggestion. We revised the relevant section based on the ODD Protocol.

4. I appreciate Table 1, but several important criteria are missing that would improve the clarity of the comparison and better highlight the features of the enhanced ABM. I recommend reviewing Table 5 in the paper https://doi.org/10.48550/arXiv.2411.04297 for reference.

Answer: We combined and extended Table 1 with the characteristics in the example paper, thank you for this suggestion. (Only the table captions is highlighted, but entire table is replaced.)

Finally, please scan the manuscript again for any remaining typos errors.

Answer: We performed another proofreading pass (we used US spelling for this manuscript), and hopefully we managed to catch all remaining typos (not highlighted).

I hope these suggestions are helpful, and I look forward to seeing the revised version.

Answer: Thank you again for the helpful suggestions!

---

## [Decision Letter · Decision Letter 2]

25 Nov 2025

Understanding the mechanisms of infodemics: equation-based vs. agent-based models

PONE-D-24-58654R2

Dear Dr. Patrascu,

We’re pleased to inform you that your manuscript has been judged scientifically suitable for publication and will be formally accepted for publication once it meets all outstanding technical requirements.

Kind regards,

Siew Ann Cheong, Ph.D.

Academic Editor

PLOS ONE

Additional Editor Comments (optional):

Reviewers' comments:

Reviewer's Responses to Questions

**Comments to the Author**

1. If the authors have adequately addressed your comments raised in a previous round of review and you feel that this manuscript is now acceptable for publication, you may indicate that here to bypass the “Comments to the Author” section, enter your conflict of interest statement in the “Confidential to Editor” section, and submit your "Accept" recommendation.

Reviewer #3: All comments have been addressed

2. Is the manuscript technically sound, and do the data support the conclusions?

Reviewer #3: Yes

3. Has the statistical analysis been performed appropriately and rigorously? 

Reviewer #3: Yes

4. Have the authors made all data underlying the findings in their manuscript fully available?

Reviewer #3: Yes

5. Is the manuscript presented in an intelligible fashion and written in standard English?

Reviewer #3: Yes

6. Review Comments to the Author

Reviewer #3: All comments have been addressed and the manuscript has been improved. I believe

can now be published

7. PLOS authors have the option to publish the peer review history of their article (what does this mean?). If published, this will include your full peer review and any attached files.

Reviewer #3: No

---

## [Editor Report · Acceptance letter]

PONE-D-24-58654R2

PLOS ONE

Dear Dr. Patrascu,

I'm pleased to inform you that your manuscript has been deemed suitable for publication in PLOS ONE. Congratulations! Your manuscript is now being handed over to our production team.

Kind regards,

on behalf of

Dr. Siew Ann Cheong

Academic Editor

PLOS ONE